# INFOBERT: IMPROVING ROBUSTNESS OF LANGUAGE MODELS FROM AN INFORMATION THEORETIC PERSPECTIVE

[*]**Boxin Wang**[1], **Shuohang Wang**[2], **Yu Cheng**[2], **Zhe Gan**[2], **Ruoxi Jia**[3], **Bo Li**[1], **Jingjing Liu**[2]

[1]University of Illinois at Urbana-Champaign  [2] Microsoft Dynamics 365 AI Research  [3] Virginia Tech

{boxinw2,lbo}@illinois.edu  {shuohang.wang,yu.cheng,zhe.gan,jingjl}@microsoft.com

## ABSTRACT

Large-scale pre-trained language models such as BERT and RoBERTa have achieved state-of-the-art performance across a wide range of NLP tasks. Recent studies, however, show that such BERT-based models are vulnerable facing the threats of textual adversarial attacks. We aim to address this problem from an information-theoretic perspective, and propose InfoBERT, a novel learning framework for robust fine-tuning of pre-trained language models. InfoBERT contains two mutual-information-based regularizers for model training: ($i$) an Information Bottleneck regularizer, which suppresses noisy mutual information between the input and the feature representation; and ($ii$) an Anchored Feature regularizer, which increases the mutual information between local stable features and global features. We provide a principled way to theoretically analyze and improve the robustness of language models in both standard and adversarial training. Extensive experiments demonstrate that InfoBERT achieves state-of-the-art robust accuracy over several adversarial datasets on Natural Language Inference (NLI) and Question Answering (QA) tasks. Our code is available at https://github.com/AI-secure/InfoBERT.

## 1 INTRODUCTION

Self-supervised representation learning pre-trains good feature extractors from massive unlabeled data, which show promising transferability to various downstream tasks. Recent success includes large-scale pre-trained language models (*e.g.,* BERT, RoBERTa, and GPT-3 (Devlin et al., 2019; Liu et al., 2019; Brown et al., 2020)), which have advanced state of the art over a wide range of NLP tasks such as NLI and QA, even surpassing human performance. Specifically, in the computer vision domain, many studies have shown that self-supervised representation learning is essentially solving the problem of maximizing the mutual information (MI) $I(X;T)$ between the input $X$ and the representation $T$ (van den Oord et al., 2018; Belghazi et al., 2018; Hjelm et al., 2019; Chen et al., 2020). Since MI is computationally intractable in high-dimensional feature space, many MI estimators (Belghazi et al., 2018) have been proposed to serve as lower bounds (Barber & Agakov, 2003; van den Oord et al., 2018) or upper bounds (Cheng et al., 2020) of MI. Recently, Kong et al. point out that the MI maximization principle of representation learning can be applied to not only computer vision but also NLP domain, and propose a unified view that recent pre-trained language models are maximizing a lower bound of MI among different segments of a word sequence.

On the other hand, deep neural networks are known to be prone to adversarial examples (Goodfellow et al., 2015; Papernot et al., 2016; Eykholt et al., 2017; Moosavi-Dezfooli et al., 2016), *i.e.,* the outputs of neural networks can be arbitrarily wrong when human-imperceptible adversarial perturbations are added to the inputs. Textual adversarial attacks typically perform word-level substitution (Ebrahimi et al., 2018; Alzantot et al., 2018; Ren et al., 2019) or sentence-level paraphrasing (Iyyer et al., 2018; Zhang et al., 2019) to achieve semantic/utility preservation that seems innocuous to human, while fools NLP models. Recent studies (Jin et al., 2020; Zang et al., 2020; Nie et al., 2020; Wang et al., 2020) further show that even large-scale pre-trained language models (LM) such as

---

[*]Work was done during Boxin Wang's Summer internship in Microsoft Dynamics 365 AI Research.

BERT are vulnerable to adversarial attacks, which raises the challenge of building robust real-world LM applications against unknown adversarial attacks.

We investigate the robustness of language models from an information theoretic perspective, and propose a novel learning framework InfoBERT, which focuses on improving the robustness of language representations by fine-tuning both local features (word-level representation) and global features (sentence-level representation) for robustness purpose. InfoBERT considers two MI-based regularizers: ($i$) the *Information Bottleneck* regularizer manages to extract approximate minimal sufficient statistics for downstream tasks, while removing excessive and noisy information that may incur adversarial attacks; ($ii$) the *Anchored Feature* regularizer carefully selects useful local stable features that are invulnerable to adversarial attacks, and maximizes the mutual information between local stable features and global features to improve the robustness of the global representation. In this paper, we provide a detailed theoretical analysis to explicate the effect of InfoBERT for robustness improvement, along with extensive empirical adversarial evaluation to validate the theory.

Our contributions are summarized as follows. ($i$) We propose a novel learning framework InfoBERT from the information theory perspective, aiming to effectively improve the robustness of language models. ($ii$) We provide a principled theoretical analysis on model robustness, and propose two MI-based regularizers to refine the local and global features, which can be applied to both standard and adversarial training for different NLP tasks. ($iii$) Comprehensive experimental results demonstrate that InfoBERT can substantially improve robust accuracy by a large margin without sacrificing the benign accuracy, yielding the state-of-the-art performance across multiple adversarial datasets on NLI and QA tasks.

## 2 RELATED WORK

**Textual Adversarial Attacks/Defenses** Most existing textual adversarial attacks focus on word-level adversarial manipulation. Ebrahimi et al. (2018) is the first to propose a whitebox gradient-based attack to search for adversarial word/character substitution. Following work (Alzantot et al., 2018; Ren et al., 2019; Zang et al., 2020; Jin et al., 2020) further constrains the perturbation search space and adopts Part-of-Speech checking to make NLP adversarial examples look natural to human.

To defend against textual adversarial attacks, existing work can be classified into three categories: ($i$) *Adversarial Training* is a practical method to defend against adversarial examples. Existing work either uses PGD-based attacks to generate adversarial examples in the embedding space of NLP as data augmentation (Zhu et al., 2020a), or regularizes the standard objective using virtual adversarial training (Jiang et al., 2020; Liu et al., 2020; Gan et al., 2020). However, one drawback is that the threat model is often unknown, which renders adversarial training less effective when facing unseen attacks. ($ii$) *Interval Bound Propagation* (IBP) (Dvijotham et al., 2018) is proposed as a new technique to consider the worst-case perturbation theoretically. Recent work (Huang et al., 2019; Jia et al., 2019) has applied IBP in the NLP domain to certify the robustness of models. However, IBP-based methods rely on strong assumptions of model architecture and are difficult to adapt to recent transformer-based language models. ($iii$) *Randomized Smoothing* (Cohen et al., 2019) provides a tight robustness guarantee in $\ell_2$ norm by smoothing the classifier with Gaussian noise. Ye et al. (2020) adapts the idea to the NLP domain, and replace the Gaussian noise with synonym words to certify the robustness as long as adversarial word substitution falls into predefined synonym sets. However, to guarantee the completeness of the synonym set is challenging.

**Representation Learning** MI maximization principle has been adopted by many studies on self-supervised representation learning (van den Oord et al., 2018; Belghazi et al., 2018; Hjelm et al., 2019; Chen et al., 2020). Specifically, InfoNCE (van den Oord et al., 2018) is used as the lower bound of MI, forming the problem as contrastive learning (Saunshi et al., 2019; Yu et al., 2020). However, Tian et al. (2020) suggests that the InfoMax (Linsker, 1988) principle may introduce excessive and noisy information, which could be adversarial. To generate robust representation, Zhu et al. (2020b) formalizes the problem from a mutual-information perspective, which essentially performs adversarial training for worst-case perturbation, while mainly considers the continuous space in computer vision. In contrast, InfoBERT originates from an information-theoretic perspective and is compatible with both standard and adversarial training for discrete input space of language models.

## 3 INFOBERT

Before diving into details, we first discuss the textual adversarial examples we consider in this paper. We mainly focus on the dominant word-level attack as the main threat model, since it achieves higher attack success and is less noticeable to human readers than other attacks. Due to the discrete nature of text input space, it is difficult to measure adversarial distortion on token level. Instead, because most word-level adversarial attacks (Li et al., 2019; Jin et al., 2020) constrain word perturbations via the bounded magnitude in the semantic embedding space, by adapting from Jacobsen et al. (2019), we define the adversarial text examples with distortions constrained in the embedding space.

**Definition 3.1.** ($\epsilon$-bounded Textual Adversarial Examples). Given a sentence $x = [x_1; x_2; ...; x_n]$, where $x_i$ is the word at the $i$-th position, the $\epsilon$-bounded adversarial sentence $x' = [x_1'; x_2'; ...; x_n']$ for a classifier $\mathcal{F}$ satisfies: (1) $\mathcal{F}(x) = o(x) = o(x')$ but $\mathcal{F}(x') \neq o(x')$, where $o(\cdot)$ is the oracle (*e.g.*, human decision-maker); (2) $||t_i - t_i'||_2 \leq \epsilon$ for $i = 1, 2, ..., n$, where $\epsilon \geq 0$ and $t_i$ is the word embedding of $x_i$.

### 3.1 INFORMATION BOTTLENECK AS A REGULARIZER

In this section, we first discuss the general IB implementation, and then explain how IB formulation is adapted to InfoBERT as a regularizer along with theoretical analysis to support why IB regularizer can help improve the robustness of language models. The IB principle formulates the goal of deep learning as an information-theoretic trade-off between representation compression and predictive power (Tishby & Zaslavsky, 2015). Given the input source $X$, a deep neural net learns the internal representation $T$ of some intermediate layer and maximizes the MI between $T$ and label $Y$, so that $T$ subject to a constraint on its complexity contains sufficient information to infer the target label $Y$. Finding an optimal representation $T$ can be formulated as the maximization of the Lagrangian

$$\mathcal{L}_{\text{IB}} = I(Y; T) - \beta I(X; T), \tag{1}$$

where $\beta > 0$ is a hyper-parameter to control the tradeoff, and $I(Y; T)$ is defined as:

$$I(Y; T) = \int p(y, t) \log \frac{p(y, t)}{p(y)p(t)} \, dy \, dt \,. \tag{2}$$

Since Eq. (2) is intractable, we instead use the lower bound from Barber & Agakov (2003):

$$I(Y; T) \geq \int p(y, t) \log q_\psi(y \mid t) \, dy \, dt \,, \tag{3}$$

where $q_\psi(y|t)$ is the variational approximation learned by a neural network parameterized by $\psi$ for the true distribution $p(y|t)$. This indicates that maximizing the lower bound of the first term of IB $I(Y; T)$ is equivalent to minimizing the task cross-entropy loss $\ell_{\text{task}} = H(Y \mid T)$.

To derive a tractable lower bound of IB, we here use an upper bound (Cheng et al., 2020) of $I(X; T)$

$$I(X; T) \leq \int p(x, t) \log(p(t \mid x)) \, dx \, dt - \int p(x)p(t) \log(p(t \mid x)) \, dx \, dt \,. \tag{4}$$

By combining Eq. (3) and (4), we can maximize the tractable lower bound $\hat{\mathcal{L}}_{\text{IB}}$ of IB in practice by:

$$\hat{\mathcal{L}}_{\text{IB}} = \frac{1}{N} \sum_{i=1}^{N} \left[ \log q_\psi(y^{(i)} \mid t^{(i)}) \right] - \frac{\beta}{N} \sum_{i=1}^{N} \left[ \log(p(t^{(i)} \mid x^{(i)})) - \frac{1}{N} \sum_{j=1}^{N} \log(p(t^{(j)} \mid x^{(i)})) \right] \tag{5}$$

with data samples $\{x^{(i)}, y^{(i)}\}_{i=1}^{N}$, where $q_\psi$ can represent any classification model (*e.g.*, BERT), and $p(t \mid x)$ can be viewed as the feature extractor $f_\theta : \mathcal{X} \rightarrow \mathcal{T}$, where $\mathcal{X}$ and $\mathcal{T}$ are the support of the input source $X$ and extracted feature $T$, respectively.

The above is a general implementation of IB objective function. In InfoBERT, we consider $T$ as the features consisting of the local word-level features after the BERT embedding layer $f_\theta$. The following BERT self-attentive layers along with the linear classification head serve as $q_\psi(y|t)$ that predicts the target $Y$ given representation $T$.

Formally, given random variables $X = [X_1; X_2; ...; X_n]$ representing input sentences with $X_i$ (word token at $i$-th index), let $T = [T_1; ...; T_n] = f_\theta([X_1; X_2; ...; X_n]) = [f_\theta(X_1); f_\theta(X_2); ...; f_\theta(X_n)]$

denote the random variables representing the features generated from input $X$ via the BERT embedding layer $f_\theta$, where $T_i \in \mathbb{R}^d$ is the high-dimensional word-level local feature for word $X_i$. Due to the high dimensionality $d$ of each word feature (*e.g.*, 1024 for BERT-large), when the sentence length $n$ increases, the dimensionality of features $T$ becomes too large to compute $I(X;T)$ in practice. Thus, we propose to maximize a localized formulation of IB $\mathcal{L}_{\mathrm{LIB}}$ defined as:

$$\mathcal{L}_{\mathrm{LIB}} := I(Y;T) - n\beta \sum_{i=1}^{n} I(X_i;T_i). \tag{6}$$

**Theorem 3.1.** *(Lower Bound of $\mathcal{L}_{IB}$) Given a sequence of random variables $X = [X_1; X_2; ...; X_n]$ and a deterministic feature extractor $f_\theta$, let $T = [T_1; ...; T_n] = [f_\theta(X_1); f_\theta(X_2); ...; f_\theta(X_n)]$. Then the localized formulation of IB $\mathcal{L}_{LIB}$ is a lower bound of $\mathcal{L}_{IB}$ (Eq. (1)), i.e.,*

$$I(Y;T) - \beta I(X;T) \geq I(Y;T) - n\beta \sum_{i=1}^{n} I(X_i;T_i). \tag{7}$$

Theorem 3.1 indicates that we can maximize the localized formulation of $\mathcal{L}_{\mathrm{LIB}}$ as a lower bound of IB $\mathcal{L}_{\mathrm{IB}}$ when $I(X;T)$ is difficult to compute. In Eq. (6), if we regard the first term ($I(Y;T)$) as a task-related objective, the second term ($-n\beta \sum_{i=1}^{n} I(X_i;T_i)$) can be considered as a regularization term to constrain the complexity of representation $T$, thus named as *Information Bottleneck regularizer*. Next, we give a theoretical analysis for the adversarial robustness of IB and demonstrate why localized IB objective function can help improve the robustness to adversarial attacks.

Following Definition 3.1, let $T = [T_1; T_2; ...; T_n]$ and $T' = [T_1'; T_2'; ...; T_n']$ denote the features for the benign sentence $X$ and adversarial sentence $X'$. The distributions of $X$ and $X'$ are denoted by probability $p(x)$ and $q(x)$ with the support $\mathcal{X}$ and $\mathcal{X}'$, respectively. We assume that the feature representation $T$ has finite support denoted by $\mathcal{T}$ considering the finite vocabulary size in NLP.

**Theorem 3.2.** *(Adversarial Robustness Bound) For random variables $X = [X_1; X_2; ...; X_n]$ and $X' = [X_1'; X_2'; ...; X_n']$, let $T = [T_1; T_2; ...; T_n] = [f_\theta(X_1); f_\theta(X_2); ...; f_\theta(X_n)]$ and $T' = [T_1'; T_2'; ...; T_n'] = [f_\theta(X_1'); f_\theta(X_2'); ...; f_\theta(X_n')]$ with finite support $\mathcal{T}$, where $f_\theta$ is a deterministic feature extractor. The performance gap between benign and adversarial data $|I(Y;T) - I(Y;T')|$ is bounded above by*

$$|I(Y;T) - I(Y;T')| \leq B_0 + B_1 \sum_{i=1}^{n} \sqrt{|\mathcal{T}|}(I(X_i;T_i))^{1/2} + B_2 \sum_{i=1}^{n} |\mathcal{T}|^{3/4}(I(X_i;T_i))^{1/4}$$

$$+ B_3 \sum_{i=1}^{n} \sqrt{|\mathcal{T}|}(I(X_i';T_i'))^{1/2} + B_4 \sum_{i=1}^{n} |\mathcal{T}|^{3/4}(I(X_i';T_i'))^{1/4}, \tag{8}$$

*where $B_0, B_1, B_2, B_3$ and $B_4$ are constants depending on the sequence length $n$, $\epsilon$ and $p(x)$.*

The sketch of the proof is to express the difference of $|I(Y;T) - I(Y';T)|$ in terms of $I(X_i;T_i)$. Specifically, Eq. (25) factorizes the difference into two summands. The first summand, the conditional entropy $|H(T \mid Y) - H(T' \mid Y)|$, can be bound by Eq. (42) in terms of MI between benign/adversarial input and representation $I(X_i;T_i)$ and $I(X_i';T_i')$. The second summand $|H(T) - H(T')|$ has a constant upper bound (Eq. (85)), since language models have bounded vocabulary size and embedding space, and thus have bounded entropy.

The intuition of Theorem 3.2 is to bound the adversarial performance drop $|I(Y;T) - I(Y;T')|$ by $I(X_i;T_i)$. As explained in Eq. (3), $I(Y;T)$ and $I(Y;T')$ can be regarded as the model performance on benign and adversarial data. Thus, the LHS of the bound represents such a performance gap. The adversarial robustness bound of Theorem 3.2 indicates that the performance gap becomes closer when $I(X_i;T_i)$ and $I(X_i';T_i')$ decrease. Note that our IB regularizer in the objective function Eq. (6) achieves the same goal of minimizing $I(X_i;T_i)$ while learning the most efficient information features, or approximate minimal sufficient statistics, for downstream tasks. Theorem 3.2 also suggests that combining adversarial training with our IB regularizer can further minimize $I(X_i';T_i')$, leading to better robustness, which is verified in §4.

## 3.2 ANCHORED FEATURE REGULARIZER

In addition to the IB regularizer that suppresses noisy information that may incur adversarial attacks, we propose a novel regularizer termed "Anchored Feature Regularizer", which extracts local

---

**Algorithm 1 - Local Anchored Feature Extraction.** This algorithm takes in the word local features and returns the index of local anchored features.

1: **Input:** Word local features $t$, upper and lower threshold $c_h$ and $c_l$
2: $\delta \leftarrow 0$     *// Initialize the perturbation vector $\delta$*
3: $g(\delta) = \nabla_\delta \ell_{\text{task}}(q_\psi(t+\delta), y)$     *// Perform adversarial attack on the embedding space*
4: Sort the magnitude of the gradient of the perturbation vector from $||g(\delta)_1||_2, ||g(\delta)_2||_2, ..., ||g(\delta)_n||_2$ into $||g(\delta)_{k_1}||_2, ||g(\delta)_{k_2}||_2, ..., ||g(\delta)_{k_n}||_2$ in ascending order, where $z_i$ corresponds to its original index.
5: **Return:** $k_i, k_{i+1}, ..., k_j$, where $c_l \leq \frac{i}{n} \leq \frac{j}{n} \leq c_h$.

---

stable features and aligns them with sentence global representations, thus improving the stability and robustness of language representations.

The goal of the *local anchored* feature extraction is to find features that carry useful and stable information for downstream tasks. Instead of directly searching for local anchored features, we start with searching for *nonrobust* and *unuseful* features. To identify local nonrobust features, we perform adversarial attacks to detect which words are prone to changes under adversarial word substitution. We consider these vulnerable words as features nonrobust to adversarial threats. Therefore, global robust sentence representations should rely less on these vulnerable statistical clues. On the other hand, by examining the adversarial perturbation on each local word feature, we can also identify words that are less useful for downstream tasks. For example, stopwords and punctuation usually carry limited information, and tend to have smaller adversarial perturbations than words containing more effective information. Although these unuseful features are barely changed under adversarial attacks, they contain insufficient information and should be discarded. After identifying the nonrobust and unuseful features, we treat the remaining local features in the sentences as useful stable features and align the global feature representation based on them.

During the local anchored feature extraction, we perform "virtual" adversarial attacks that generate adversarial perturbation in the embedding space, as it abstracts the general idea for existing word-level adversarial attacks. Formally, given an input sentence $x = [x_1; x_2; ...; x_n]$ with its corresponding local embedding representation $t = [t_1; ...; t_n]$, where $x$ and $t$ are the realization of random variables $X$ and $T$, we generate adversarial perturbation $\delta$ in the embedding space so that the task loss $\ell_{\text{task}}$ increases. The adversarial perturbation $\delta$ is initialized to zero, and the gradient of the loss with respect to $\delta$ is calculated by $g(\delta) = \nabla_\delta \ell_{\text{task}}(q_\psi(t+\delta), y)$ to update $\delta \leftarrow \prod_{||\delta||_F \leq \epsilon}(\eta g(\delta)/||g(\delta)||_F)$. The above process is similar to one-step PGD with zero-initialized perturbation $\delta$. Since we only care about the ranking of perturbation to decide on robust features, in practice we skip the update of $\delta$ to save computational cost, and simply examine the $\ell_2$ norm of the gradient $g(\delta)_i$ of the perturbation on each word feature $t_i$. A feasible plan is to choose the words whose perturbation is neither too large (nonrobust features) nor too small (unuseful features), *e.g.*, the words whose perturbation rankings are among $50\% \sim 80\%$ of all the words. The detailed procedures are provided in Algorithm 1.

After local anchored features are extracted, we propose to align sentence global representations $Z$ with our local anchored features $T_i$. In practice, we can use the final-layer [CLS] embedding to represent global sentence-level feature $Z$. Specifically, we use the information theoretic tool to increase the mutual information $I(T_i; Z)$ between local anchored features $T_i$ and sentence global representations $Z$, so that the global representations can share more robust and useful information with the local anchored features and focus less on the nonrobust and unuseful ones. By incorporating the term $I(T_i; Z)$ into the previous objective function Eq. (6), our final objective function becomes:

$$\max \ I(Y; T) - n\beta \sum_{i=1}^{n} I(X_i; T_i) + \alpha \sum_{j=1}^{M} I(T_{k_j}; Z), \tag{9}$$

where $T_{k_j}$ are the local anchored features selected by Algorithm 1 and $M$ is the number of local anchored features. An illustrative figure can be found in Appendix Figure 2.

In addition, due to the intractability of computing MI, we use InfoNCE (van den Oord et al., 2018) as the lower bound of MI to approximate the last term $I(T_{k_j}; Z)$:

$$\hat{I}^{(\text{InfoNCE})}(T_i; Z) := \mathbb{E}_{\mathbb{P}}\left[g_\omega(t_i, z) - \mathbb{E}_{\tilde{\mathbb{P}}}\left[\log \sum_{t_i'} e^{g_\omega(t_i', z)}\right]\right], \tag{10}$$

where $g_\omega(\cdot, \cdot)$ is a score function (or critic function) approximated by a neural network, $t_i$ are the positive samples drawn from the joint distribution $\mathbb{P}$ of local anchored features and global representations, and $t_i'$ are the negative samples drawn from the distribution of nonrobust and unuseful features $\tilde{\mathbb{P}}$.

## 4 EXPERIMENTS

In this section, we demonstrate how effective InfoBERT improves the robustness of language models over multiple NLP tasks such as NLI and QA. We evaluate InfoBERT against both strong adversarial datasets and state-of-the-art adversarial attacks.

### 4.1 EXPERIMENTAL SETUP

**Adversarial Datasets** The following adversarial datasets and adversarial attacks are used to evaluate the robustness of InfoBERT and baselines. **(I)** Adversarial NLI (ANLI) (Nie et al., 2020) is a large-scale NLI benchmark, collected via an iterative, adversarial, human-and-model-in-the-loop procedure to attack BERT and RoBERTa. ANLI dataset is a strong adversarial dataset which can easily reduce the accuracy of BERT$_{\text{Large}}$ to $0\%$. **(II)** Adversarial SQuAD (Jia & Liang, 2017) dataset is an adversarial QA benchmark dataset generated by a set of handcrafted rules and refined by crowdsourcing. Since adversarial training data is not provided, we fine-tune RoBERTa$_{\text{Large}}$ on benign SQuAD training data (Rajpurkar et al., 2016) only, and test the models on both benign and adversarial test sets. **(III)** TextFooler (Jin et al., 2020) is the state-of-the-art word-level adversarial attack method to generate adversarial examples. To create an adversarial evaluation dataset, we sampled $1,000$ examples from the test sets of SNLI and MNLI respectively, and run TextFooler against BERT$_{\text{Large}}$ and RoBERTa$_{\text{Large}}$ to obtain the adversarial text examples.

**Baselines** Since IBP-based methods (Huang et al., 2019; Jia et al., 2019) cannot be applied to large-scale language models yet, and the randomized-smoothing-based method (Ye et al., 2020) achieves limited certified robustness, we compare InfoBERT against three competitive baselines based on adversarial training: **(I)** FreeLB (Zhu et al., 2020a) applies adversarial training to language models during fine-tuning stage to improve generalization. In §4.2, we observe that FreeLB can boost the robustness of language models by a large margin. **(II)** SMART (Jiang et al., 2020) uses adversarial training as smoothness-inducing regularization and Bregman proximal point optimization during fine-tuning, to improve the generalization and robustness of language models. **(III)** ALUM (Liu et al., 2020) performs adversarial training in both pre-training and fine-tuning stages, which achieves substantial performance gain on a wide range of NLP tasks. Due to the high computational cost of adversarial training, we compare InfoBERT to ALUM and SMART with the best results reported in the original papers.

**Evaluation Metrics** We use robust accuracy or robust F1 score to measure how robust the baseline models and InfoBERT are when facing adversarial data. Specifically, robust accuracy is calculated by: $\text{Acc} = \frac{1}{|D_{\text{adv}}|} \sum_{\boldsymbol{x}' \in D_{\text{adv}}} \mathbb{1}[\arg\max q_\psi(f_\theta(x')) \equiv y]$, where $D_{\text{adv}}$ is the adversarial dataset, $y$ is the ground-truth label, $\arg\max$ selects the class with the highest logits and $\mathbb{1}(\cdot)$ is the indicator function. Similarly, robust F1 score is calculated by: $\text{F1} = \frac{1}{|D_{\text{adv}}|} \sum_{\boldsymbol{x}' \in D_{\text{adv}}} v(\arg\max q_\psi(f_\theta(x')), a)$, where $v(\cdot, \cdot)$ is the F1 score between the true answer $a$ and the predicted answer $\arg\max q_\psi(f_\theta(x'))$, and $\arg\max$ selects the answer with the highest probability (see Rajpurkar et al. (2016) for details).

**Implementation Details** To demonstrate InfoBERT is effective for different language models, we apply InfoBERT to both pretrained RoBERTa$_{\text{Large}}$ and BERT$_{\text{Large}}$. Since InfoBERT can be applied to both standard training and adversarial training, we here use FreeLB as the adversarial training implementation. InfoBERT is fine-tuned for 2 epochs for the QA task, and 3 epochs for the NLI task. More implementation details such as $\alpha, \beta, c_h, c_l$ selection can be found in Appendix A.1.

| Training | Model | Method | Dev | | | | Test | | | |
|---|---|---|---|---|---|---|---|---|---|---|
| | | | A1 | A2 | A3 | ANLI | A1 | A2 | A3 | ANLI |
| Standard Training | RoBERTa | Vanilla | 49.1 | 26.5 | 27.2 | 33.8 | 49.2 | 27.6 | 24.8 | 33.2 |
| | | InfoBERT | 47.8 | 31.2 | 31.8 | **36.6** | 47.3 | 31.2 | 31.1 | **36.2** |
| | BERT | Vanilla | 20.7 | 26.9 | 31.2 | 26.6 | 21.8 | 28.3 | 28.8 | 26.5 |
| | | InfoBERT | 26.0 | 30.1 | 31.2 | **29.2** | 26.4 | 29.7 | 29.8 | **28.7** |
| Adversarial Training | RoBERTa | FreeLB | 50.4 | 28.0 | 28.5 | 35.2 | 48.1 | 30.4 | 26.3 | 34.4 |
| | | InfoBERT | 48.4 | 29.3 | 31.3 | **36.0** | 50.0 | 30.6 | 29.3 | **36.2** |
| | BERT | FreeLB | 23.0 | 29.0 | 32.2 | 28.3 | 22.2 | 28.5 | 30.8 | 27.4 |
| | | InfoBERT | 28.3 | 30.2 | 33.8 | **30.9** | 25.9 | 28.1 | 30.3 | **28.2** |

Table 1: Robust accuracy on the ANLI dataset. Models are trained on the benign datasets (MNLI + SNLI) only. 'A1-A3' refers to the rounds with increasing difficulty. 'ANLI' refers to A1+A2+A3.

| Training | Model | Method | Dev | | | | Test | | | |
|---|---|---|---|---|---|---|---|---|---|---|
| | | | A1 | A2 | A3 | ANLI | A1 | A2 | A3 | ANLI |
| Standard Training | RoBERTa | Vanilla | 74.1 | 50.8 | 43.9 | 55.5 | 73.8 | 48.9 | 44.4 | 53.7 |
| | | InfoBERT | 75.2 | 49.6 | 47.8 | **56.9** | 73.9 | 50.8 | 48.8 | **57.3** |
| | BERT | Vanilla | 58.5 | 46.1 | 45.5 | 49.8 | 57.4 | 48.3 | 43.5 | 49.3 |
| | | InfoBERT | 59.3 | 48.9 | 45.5 | **50.9** | 60.0 | 46.9 | 44.8 | **50.2** |
| Adversarial Training | RoBERTa | FreeLB | 75.2 | 47.4 | 45.3 | 55.3 | 73.3 | 50.5 | 46.8 | 56.2 |
| | | SMART | 74.5 | 50.9 | 47.6 | 57.1 | 72.4 | 49.8 | 50.3 | 57.1 |
| | | ALUM | 73.3 | 53.4 | 48.2 | 57.7 | 72.3 | 52.1 | 48.4 | 57.0 |
| | | InfoBERT | 76.4 | 51.7 | 48.6 | **58.3** | 75.5 | 51.4 | 49.8 | **58.3** |
| | BERT | FreeLB | 60.3 | 47.1 | 46.3 | 50.9 | 60.3 | 46.8 | 44.8 | 50.2 |
| | | ALUM | 62.0 | 48.6 | 48.1 | **52.6** | 61.3 | 45.9 | 44.3 | 50.1 |
| | | InfoBERT | 60.8 | 48.7 | 45.9 | 51.4 | 63.3 | 48.7 | 43.2 | **51.2** |

Table 2: Robust accuracy on the ANLI dataset. Models are trained on both adversarial and benign datasets (ANLI (training) + FeverNLI + MNLI + SNLI).

## 4.2 EXPERIMENTAL RESULTS

**Evaluation on ANLI** As ANLI provides an adversarial training dataset, we evaluate models in two settings: 1) training models on benign data (MNLI (Williams et al., 2018) + SNLI (Bowman et al., 2015)) only, which is the case when the adversarial threat model is unknown; 2) training models on both benign and adversarial training data (SNLI+MNLI+ANLI+FeverNLI), which assumes the threat model is known in advance.

Results of the first setting are summarized in Table 1. The vanilla RoBERTa and BERT models perform poorly on the adversarial dataset. In particular, vanilla BERT$_{Large}$ with standard training achieves the lowest robust accuracy of 26.5% among all the models. We also evaluate the robustness improvement by performing adversarial training during fine-tuning, and observe that adversarial training for language models can improve not only generalization but also robustness. In contrast, InfoBERT substantially improves robust accuracy in both standard and adversarial training. The robust accuracy of InfoBERT through standard training is even higher than the adversarial training baseline FreeLB for both RoBERTa and BERT, while the training time of InfoBERT is $1/3 \sim 1/2$ less than FreeLB. This is mainly because FreeLB requires multiple steps of PGD attacks to generate adversarial examples, while InfoBERT essentially needs only 1-step PGD attack for anchored feature selection.

Results of the second setting are provided in Table 2, which shows InfoBERT can further improve robust accuracy for both standard and adversarial training. Specifically, when combined with adversarial training, InfoBERT achieves the state-of-the-art robust accuracy of 58.3%, outperforming all existing baselines. Note that although ALUM achieves higher accuracy for BERT on the dev set, it tends to overfit on the dev set, therefore performing worse than InfoBERT on the test set.

| Training | Model | Method | SNLI | MNLI (m/mm) | adv-SNLI (BERT) | adv-MNLI (BERT) | adv-SNLI (RoBERTa) | adv-MNLI (RoBERTa) |
|---|---|---|---|---|---|---|---|---|
| Standard Training | RoBERTa | Vanilla | 92.6 | **90.8/90.6** | 56.6 | 68.1/68.6 | 19.4 | 24.9/24.9 |
| | | InfoBERT | **93.3** | 90.5/90.4 | **59.8** | **69.8/70.6** | **42.5** | **50.3/52.1** |
| | BERT | Vanilla | 91.3 | **86.7/86.4** | 0.0 | 0.0/0.0 | 44.9 | 57.0/57.5 |
| | | InfoBERT | **91.7** | 86.2/86.0 | **36.7** | **43.5/46.6** | **45.4** | **57.2/58.6** |
| Adversarial Training | RoBERTa | FreeLB | **93.4** | 90.1/90.3 | 60.4 | 70.3/72.1 | 41.2 | 49.5/50.6 |
| | | InfoBERT | 93.1 | **90.7/90.4** | **62.3** | **73.2/73.1** | **43.4** | **56.9/55.5** |
| | BERT | FreeLB | **92.4** | 86.9/86.5 | 46.6 | 60.0/60.7 | 50.5 | 64.0/62.9 |
| | | InfoBERT | 92.2 | **87.2/87.2** | **50.8** | **61.3/62.7** | **52.6** | **65.6/67.3** |

Table 3: Robust accuracy on the adversarial SNLI and MNLI(-m/mm) datasets generated by TextFooler based on blackbox BERT/RoBERTa (denoted in brackets of the header). Models are trained on the benign datasets (MNLI+SNLI) only.

| Training | Method | benign | AddSent | AddOneSent |
|---|---|---|---|---|
| Standard Training | Vanilla | **93.5**/86.9 | 72.9/66.6 | 80.6/74.3 |
| | InfoBERT | **93.5**/87.0 | **78.5/72.9** | **84.6/78.3** |
| Adversarial Training | FreeLB | **93.8/87.3** | 76.3/70.3 | 82.3/76.2 |
| | ALUM | - | 75.5/69.4 | 81.4/75.9 |
| | InfoBERT | 93.7/87.0 | **78.0/71.8** | **83.6/77.1** |

Table 4: Robust F1/EM scores based on RoBERTa$_{\text{Large}}$ on the adversarial SQuAD datasets (AddSent and AddOne-Sent). Models are trained on standard SQuAD 1.0 dataset.

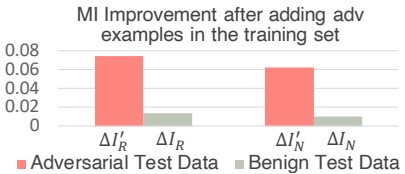

MI Improvement after adding adv examples in the training set

- Adversarial Test Data  - Benign Test Data

Figure 1: Local anchored features contribute more to MI improvement than nonrobust/unuseful features, unveiling closer relation with robustness.

**Evaluation against TextFooler** InfoBERT can defend against not only human-crafted adversarial examples (*e.g.*, ANLI) but also those generated by adversarial attacks (*e.g.*, TextFooler). Results are summarized in Table 3. We can see that InfoBERT barely affects model performance on the benign test data, and in the case of adversarial training, InfoBERT even boosts the benign test accuracy. Under the TextFooler attack, the robust accuracy of the vanilla BERT drops to $0.0\%$ on both MNLI and SNLI datasets, while RoBERTa drops from $90\%$ to around $20\%$. We observe that both adversarial training and InfoBERT with standard training can improve robust accuracy by a comparable large margin, while InfoBERT with adversarial training achieves the best performance among all models, confirming the hypothesis in Theorem 3.2 that combining adversarial training with IB regularizer can further minimize $I(X_i'; T_i')$, leading to better robustness than the vanilla one.

**Evaluation on Adversarial SQuAD** Previous experiments show that InfoBERT can improve model robustness for NLI tasks. Now we demonstrate that InfoBERT can also be adapted to other NLP tasks such as QA in Table 4. Similar to our observation on NLI dataset, we find that InfoBERT barely hurts the performance on the benign test data, and even improves it in some cases. Moreover, InfoBERT substantially improves model robustness when presented with adversarial QA test sets (AddSent and AddOneSent). While adversarial training does help improve robustness, InfoBERT can further boost the robust performance by a larger margin. In particular, InfoBERT through standard training achieves the state-of-the-art robust F1/EM score as $78.5/72.9$ compared to existing adversarial training baselines, and in the meantime requires only half the training time of adversarial-training-based methods.

### 4.3 ANALYSIS OF LOCAL ANCHORED FEATURES

We conduct an ablation study to further validate that our anchored feature regularizer indeed filters out nonrobust/unuseful information. As shown in Table 1 and 2, adding adversarial data in the training set can significantly improve model robustness. To find out what helps improve the robustness from the MI perspective, we first calculate the MI between anchored features and global features $\frac{1}{M} \sum_{j=1}^{M} I(T_{k_j}; Z)$ on the adversarial test data and benign test data, based on the model trained without adversarial training data (denoted by $I_\text{R}'$ and $I_\text{R}$). We then calculate the MI between nonrobust/unuseful features and global features $\frac{1}{M'} \sum_{i=1}^{M'} I(T_{k_i}; Z)$ on the adversarial test data and

benign data as well (denoted by $I'_N$ and $I_N$). After adding adversarial examples into the training set and re-training the model, we find that the MI between the local features and the global features substantially increases on the adversarial test data, which accounts for the robustness improvement. We also observe that those local anchored features extracted by our anchored feature regularizer, as expected, contribute more to the MI improvement. As shown in Figure 1, the MI improvement of anchored features on adversarial test data $\Delta I'_R$ (red bar on the left) is higher than that of nonrobust/unuseful $\Delta I'_N$ (red bar on the right), thus confirming that local anchored features discovered by our anchored feature regularizer have a stronger impact on robustness than nonrobust/unuseful ones.

We conduct more ablation studies in Appendix §A.2, including analyzing the individual impact of two regularizers, the difference between global and local features for IB regularizer, hyper-parameter selection strategy and so on.

## 5    CONCLUSION

In this paper, we propose a novel learning framework InfoBERT from an information theoretic perspective to perform robust fine-tuning over pre-trained language models. Specifically, InfoBERT consists of two novel regularizers to improve the robustness of the learned representations: (a) Information Bottleneck Regularizer, learning to extract the approximated minimal sufficient statistics and denoise the excessive spurious features, and (b) Local Anchored Feature Regularizer, which improves the robustness of global features by aligning them with local anchored features. Supported by our theoretical analysis, InfoBERT provides a principled way to improve the robustness of BERT and RoBERTa against strong adversarial attacks over a variety of NLP tasks, including NLI and QA tasks. Comprehensive experiments demonstrate that InfoBERT outperforms existing baseline methods and achieves new state of the art on different adversarial datasets. We believe this work will shed light on future research directions towards improving the robustness of representation learning for language models.

## 6    ACKNOWLEDGEMENT

We gratefully thank the anonymous reviewers and meta-reviewers for their constructive feedback. We also thank Julia Hockenmaier, Alexander Schwing, Sanmi Koyejo, Fan Wu, Wei Wang, Pengyu Cheng, and many others for the helpful discussion. This work is partially supported by NSF grant No.1910100, DARPA QED-RML-FP-003, and the Intel RSA 2020.

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

# A APPENDIX

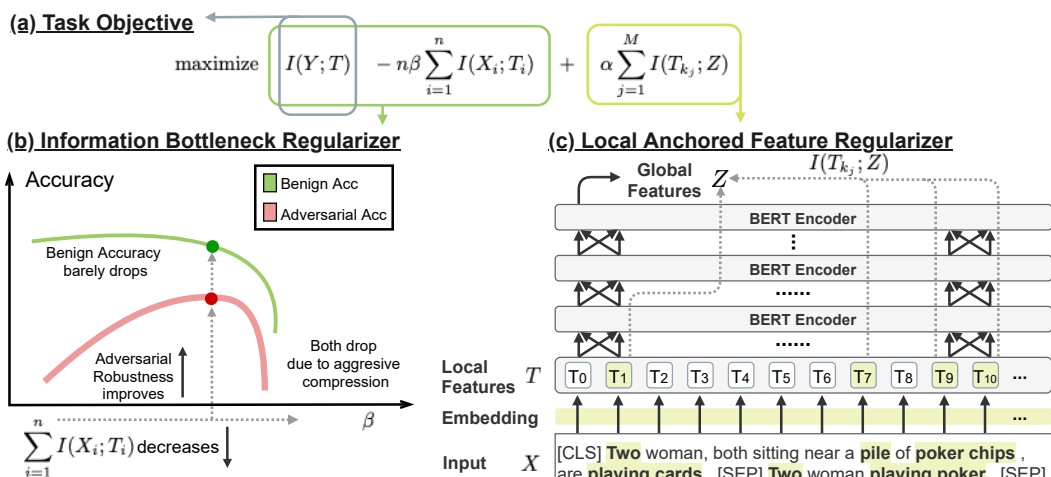

Figure 2: The complete objective function of InfoBERT, which can be decomposed into (a) standard task objective, (b) Information Bottleneck Regularizer, and (c) Local Anchored Feature Regularizer. For (b), we both theoretically and empirically demonstrate that we can improve the adversarial robustness by decreasing the mutual information of $I(X_i; T_i)$ without affecting the benign accuracy much. For (c), we propose to align the local anchored features $T_{k_j}$ (highlighted in Yellow) with the global feature $Z$ by maximizing their mutual information $I(T_{k_j}; Z)$.

## A.1 IMPLEMENTATION DETAILS

**Model Details**[1]  BERT is a transformer (Vaswani et al., 2017) based model, which is unsupervised pretrained on large corpora. We use BERT$_{\text{Large}}$-uncased as the baseline model, which has 24 layers, 1024 hidden units, 16 self-attention heads, and 340M parameters. RoBERTa$_{\text{Large}}$ shares the same architecture as BERT, but modifies key hyperparameters, removes the next-sentence pretraining objective and trains with much larger mini-batches and learning rates, which results in higher performance than BERT model on GLUE, RACE and SQuAD.

**Standard Training Details**  For both standard and adversarial training, we fine-tune InfoBERT for 2 epochs on the QA task, and for 3 epochs on the NLI task. The best model is selected based on the performance on the development set. All fine-tuning experiments are run on Nvidia V100 GPUs. For NLI task, we set the batch size to 256, learning rate to $2 \times 10^{-5}$, max sequence length to 128 and warm-up steps to 1000. For QA task, we set the batch size to 32, learning rate to $3 \times 10^{-5}$ and max sequence length to 384 without warm-up steps.

**Adversarial Training Details**[2]  Adversarial training introduces hyper-parameters including adversarial learning rates, number of PGD steps, and adversarial norm. When combing adversarial training with InfoBERT, we use FreeLB as the adversarial training implementation, and set adversarial learning rate to $10^{-1}$ or $4*10^{-2}$, adversarial steps to 3, maximal perturbation norm to $3*10^{-1}$ or $2*10^{-1}$ and initial random perturbation norm to $10^{-1}$ or 0.

**Information Bottleneck Regularizer Details**  For information bottleneck, there are different ways to model $p(t \mid x)$:

1. **Assume that $p(t \mid x)$ is unknown.** We use a neural net parameterized by $q_\theta(t \mid x)$ to learn the conditional distribution $p(t \mid x)$. We assume the distribution is a Gaussian distribution. The

---

[1]We use the huggingface implementation `https://github.com/huggingface/transformers` for BERT and RoBERTa.

[2]We follow the FreeLB implementations in `https://github.com/zhuchen03/FreeLB`.

neural net $q_\theta$ will learn the mean and variance of the Gaussian given input $x$ and representation $t$. By reparameterization trick, the neural net can be backpropagated to approximate the distribution given the training samples.

2. $p(t \mid x)$ **is known.** Since $t$ is the representation encoded by BERT, we actually already know the distribution of $p$. We also denote it as $q_\theta$, where $\theta$ is the parameter of the BERT encoder $f_\theta$. If we assume the conditional distribution is a Gaussian $\mathcal{N}(t_i, \sigma)$ for input $x_i$ whose mean is the BERT representation $t_i$ and variance is a fixed constant $\sigma$, the Eq.6 becomes

$$\mathcal{L}_{\text{LIB}} = \frac{1}{N} \sum_{i=1}^{N} \left( \left[ \log q_\psi(y^{(i)} \mid t^{(i)}) \right] - \beta \sum_{k=1}^{n} \left[ -c(\sigma)||t_k'^{(i)} - t_k^{(i)}||_2^2 + \frac{1}{n} \sum_{j=1}^{N} c(\sigma)||t_j - t_k||_2^2 \right] \right),$$
(11)

where $c(\sigma)$ is a positive constant related to $\sigma$. In practice, the sample $t_i'$ from the conditional distribution Gaussian $\mathcal{N}(t_i, \sigma)$ can be $t_i$ with some Gaussian noise, an adversarial examples of $t_i$, or $t_i$ itself (assume $\sigma = 0$).

We use the second way to model $p(t \mid x)$ for InfoBERT finally, as it gives higher robustness improvement than the first way empirically (shown in the following §A.2). We suspect that the main reason is because the first way needs to approximate the distribution $p(t \mid x)$ via another neural net which could present some difficulty in model training.

Information Bottleneck Regularizer also introduces another parameter $\beta$ to tune the trad-off between representation compression $I(X_i; T_i)$ and predictive power $I(Y; T)$. We search for the optimal $\beta$ via grid search, and set $\beta = 5 \times 10^{-2}$ for RoBERTa, $\beta = 10^{-3}$ for BERT on the NLI task. On the QA task, we set $\beta = 5 \times 10^{-5}$, which is substantially lower than $\beta$ on NLI tasks, thus containing more word-level features. We think it is mainly because the QA task relies more on the word-level representation to predict the exact answer spans. More ablation results can be found in the following §A.2.

**Anchored Feature Regularizer Details**   Anchored Feature Regularizer uses $\alpha$ to weigh the balance between predictive power and importance of anchored feature. We set $\alpha = 5 \times 10^{-3}$ for both NLI and QA tasks. Anchored Feature Regularizer also introduces upper and lower threshold $c_l$ and $c_h$ for anchored feature extraction. We set $c_h = 0.9$ and $c_l = 0.5$ for the NLI task, and set $c_h = 0.95$ and $c_l = 0.75$ for the QA task. The neural MI estimator used by infoNCE uses two-layer fully connected layer to estimate the MI with the intermediate layer hidden size set to 300.

## A.2   Additional Experimental Results

### A.2.1   Ablation Study on Information Bottleneck Regularizer

**Modeling** $p(t \mid x)$   As discussed in §A.1, we have two ways to model $p(t \mid x)$: $(i)$ using an auxiliary neural network to approximate the distribution; $(ii)$ directly using the BERT encoder $f_\theta$ to calculate the $p(t \mid x)$. Thus we implemented these two methods and compare the robustness improvement in Table 5. To eliminate other factors such as Anchored Feature Regularizer and adversarial training, we set $\alpha = 0, \beta = 5 \times 10^{-2}$ and conduct the following ablation experiments via standard training on standard datasets. We observe that although both modeling methods can improve the model robustness, modeling as BERT encoder gives a larger margin than the Auxiliary Net. Moreover, the second way barely sacrifices the performance on benign data, while the first way can hurt the benign accuracy a little bit. Therefore, we use the BERT Encoder $f_\theta$ to model the $p(t \mid x)$ in our main paper.

**Local Features v.s. Global Features**   Information Bottlenck Regularizer improves model robustness by reducing $I(X; T)$. In the main paper, we use $T$ as word-level local features. Here we consider $T$ as sentence-level global features, and compare the robustness improvement with $T$ as local features. To eliminate other factors such as Anchored Feature Regularizer and adversarial training, we set $\alpha = 0, \beta = 5 \times 10^{-2}$ and conduct the following ablation experiments via standard training.

| Model | Datasets | Method | Adversarial Accuracy (ANLI) | Benign Accuracy (MNLI/SNLI) |
|---|---|---|---|---|
| BERT | Standard Datasets | Vanilla | 26.5 | **86.7**/91.3 |
| | | Auxiliary Net | 27.1 | 83.1/90.7 |
| | | BERT Encoder $f_\theta$ | **27.7** | 85.9/**91.7** |

Table 5: Robust accuracy on the ANLI dataset. Here we refer "Standard Datasets" as training on the benign datasets (MNLI + SNLI) only. "Vanilla" refers to the vanilla BERT trained without Information Bottleneck Regularizer.

The experimental results are summarized in Table 6. We can see that while both features can boost the model robustness, using local features yield higher robust accuracy improvement than global features, especially when adversarial training dataset is added.

**Hyper-parameter Search** We perform grid search to find out the optimal $\beta$ so that the optimal trade-off between representation compression ("minimality") and predictive power ("sufficiency") is achieved. An example to search for the optimal $\beta$ on QA dataset is shown in Fingure 3, which illustrates how $\beta$ affects the F1 score on benign and adversarial datasets. We can see that from a very small $\beta$, both the robust and benign F1 scores increase, demonstrating InfoBERT can improve both robustness and generalization to some extent. When we set $\beta = 5 \times 10^{-5}$ ($\log(\beta) = -9.9$), InfoBERT achieves the best benign and adversarial accuracy. When we set a larger $\beta$ to further minimize $I(X_i; T_i)$, we observe that the benign F1 score starts to drop, indicating the increasingly compressed representation could start to hurt its predictive capability.

### A.2.2 Ablation Study on Anchored Feature Regularizer

**Visualization of Anchored Words** To explore which local anchored features are extracted, we conduct another ablation study to visualize the local anchored words. We follow the best hyper-parameters of Anchored Feature Regularizer introduced in §A.1, use the best BERT model trained on benign datasets (MNLI + SNLI) only and test on the ANLI dev set. We visualize the local anchored words in Table 7 as follows. In the first example, we find that Anchored Features mainly focus on the important features such as quantity number "Two", the verb "playing" and objects "card"/"poker" to make a robust prediction. In the second example, the matching robust features between hypothesis and premise, such as "people", "roller" v.s. "park", "flipped upside" v.s. "ride", are aligned to infer the relationship of hypothesis and premise. These anchored feature examples confirm that Anchored Feature Regularizer is able to find out useful and stable features to improve the robustness of global representation.

| Model | Datasets | Features | Adversarial Accuracy (ANLI) | Benign Accuracy (MNLI/SNLI) |
|---|---|---|---|---|
| RoBERTa | Standard Datasets | Vanilla | 33.2 | 90.8/92.6 |
| | | Global Feature | 33.8 | 90.4/93.5 |
| | | Local Feature | **33.9** | **90.6/93.7** |
| | Standard and Adversarial Datasets | Vanilla | 53.7 | **91.0**/92.6 |
| | | Global Feature | 55.1 | 90.8/**93.3** |
| | | Local Feature | **56.2** | 90.5/**93.3** |

Table 6: Robust accuracy on the ANLI dataset. Here we refer "Standard Datasets" as training on the benign datasets (MNLI + SNLI) only, and "Standard and Adversarial Datasaets" as training on the both benign and adversarial datasets (ANLI(trianing) + MNLI + SNLI + FeverNLI). "Vanilla" refers to the vanilla RoBERTa trained without Information Bottleneck Regularizer.

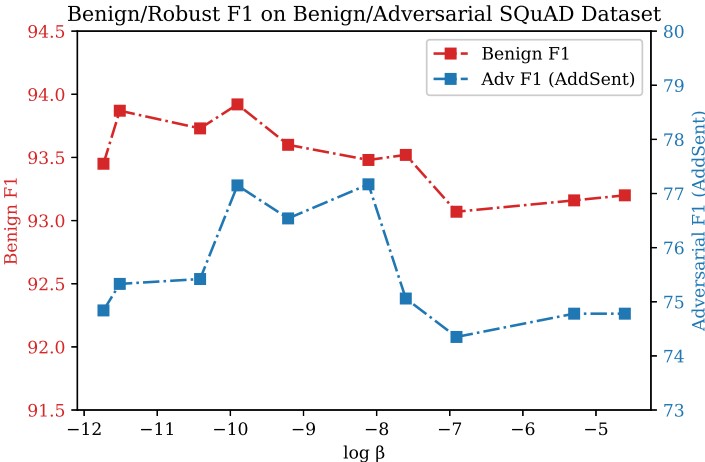

Figure 3: Benign/robust F1 score on benign/adversarial QA datasets. Models are trained on the benign SQuAD dataset with different $\beta$.

| Input (**bold** = local stable words for local anchored features.) |
| --- |
| **Premise: Two** woman, both sitting near a **pile** of **poker chips**, are **playing cards**. 
 **Hypothesis: Two** woman **playing poker**. |
| **Premise: People** are **flipped upside** - down on a bright yellow **roller** coaster. 
 **Hypothesis: People on** on a **ride** at an amusement **park**. |

Table 7: Local anchored features extracted by Anchored Feature Regularizer.

### A.2.3    ABLATION STUDY ON DISENTANGLING TWO REGULARIZERS

To understand how two regularizers contribute to the improvement of robustness separetely, we apply two regularizers individually to both the standard training and adversarial training. We refer InfoBERT trained with IB regularizer only as "InfoBERT (IBR only)" and InfoBERT trained with Anchored Feature Regularizer only as "InfoBERT (AFR only)". "InfoBERT (Both)" is the standard setting for InfoBERT, where we incorporate both regularizers during training. For "InfoBERT (IBR only)", we set $\alpha = 0$ and perform grid search to find the optimal $\beta = 5 \times 10^{-2}$. Similarly for "InfoBERT (AFR only)", we set $\beta = 0$ and find the optimal parameters as $\alpha = 5 \times 10^{-3}, c_h = 0.9$ and $c_l = 0.5$.

The results are shown in Table 8. We can see that both regularizers improve the robust accuracy on top of vanilla and FreeLB to a similar margin. Applying one of the regularizer can achieve similar performance of FreeLB, but the training time of InfoBERT is only $1/3 1/2$ less than FreeLB. Moreover, after combining both regularizers, we observe that InfoBERT achieves the best robust accuracy.

### A.2.4    EXAMPLES OF ADVERSARIAL DATASETS GENERATED BY TEXTFOOLER

We show some adversarial examples generated by TextFooler in Table 9. We can see most adversarial examples are of high quality and look valid to human while attacking the NLP models, thus confirming our adversarial dastasets created by TextFooler is a strong benchmark dataset to evaluate model robustness. However, as also noted in Jin et al. (2020), we observe that some adversarial examples look invalid to human For example, in the last example of Table 9, TextFooler replaces "stand" with "position", losing the critical information that "girls are **standing** instead of kneeling" and fooling both human and NLP models. Therefore, we expect that InfoBERT should achieve better robustness when we eliminate such invalid adversarial examples during evaluation.

| Model | Training | Method | Adversarial Accuracy (ANLI) | Benign Accuracy (MNLI/SNLI) |
|---|---|---|---|---|
| BERT | Standard Training | Vanilla | 26.5 | 86.7/91.3 |
| | | InfoBERT (IBR only) | 27.7 | 85.9/91.7 |
| | | InfoBERT (AFR only) | 28.0 | 86.6/91.9 |
| | | InfoBERT (Both) | 29.2 | 85.9/91.6 |
| | Adversarial Training | FreeLB | 27.7 | 86.7/**92.3** |
| | | InfoBERT (IBR only) | 29.3 | 87.0/**92.3** |
| | | InfoBERT (AFR only) | 30.3 | 86.9/**92.3** |
| | | InfoBERT (Both) | **30.9** | **87.2**/92.2 |

Table 8: Robust accuracy on the ANLI dataset. Models are trained on the benign datasets (MNLI + SNLI). Here we refer "IBR only" as training with Information Bottleneck Regularizer only. "AFR only" refers to InfoBERT trained with Anchored Feature Regularizer only. "Both" is the standard InfoBERT that applies two regularizers together.

---

| Input (red = Modified words, **bold** = original words.) |
|---|

***Valid Adversarial Examples***

**Premise:** A young boy is playing in the sandy water.
**Original Hypothesis:** There is a **boy** in the water.
**Adversarial Hypothesis:** There is a man in the water.

**Model Prediction:** Entailment → Contradiction

---

**Premise:** A black and brown dog is playing with a brown and white dog .
**Original Hypothesis:** Two dogs **play**.
**Adversarial Hypothesis:** Two dogs gaming.

**Model Prediction:** Entailment → Neutral

---

**Premise:** Adults and children share in the looking at something, and some young ladies stand to the side.
**Original Hypothesis:** Some children are **sleeping**.
**Adversarial Hypothesis:** Some children are dreaming.

**Model Prediction:** Contradiction → Neutral

---

**Premise:** Families with strollers waiting in front of a carousel.
**Original Hypothesis:** Families have some **dogs** in front of a carousel.
**Adversarial Hypothesis:** Families have some doggie in front of a carousel.

**Model Prediction:** Contradiction → Entailment

---

***Invalid Adversarial Examples***

**Premise:** Two girls are kneeling on the ground.
**Original Hypothesis:** Two girls **stand** around the vending **machines**.
**Adversarial Hypothesis:** Two girls position around the vending machinery.

**Model Prediction:** Contradiction → Neutral

---

Table 9: Adversarial Examples Generated by TextFooler for BERT$_{\text{Large}}$ on SNLI dataset.

### A.3 PROOFS

#### A.3.1 PROOF OF THEOREM 3.1

We first state two lemmas.

**Lemma A.1.** Given a sequence of random variables $X_1, X_2, ..., X_n$ and a deterministic function $f$, then $\forall\, i, j = 1, 2, ..., n$, we have

$$I(X_i; f(X_i)) \geq I(X_j; f(X_i)) \tag{12}$$

*Proof.* By the definition,

$$I(X_i; f(X_i)) = H(f(X_i)) - H(f(X_i) \mid X_i) \tag{13}$$
$$I(X_j; f(X_i)) = H(f(X_i)) - H(f(X_i) \mid X_j) \tag{14}$$

Since $f$ is a deterministic function,

$$H(f(X_i) \mid X_i) = 0 \tag{15}$$
$$H(f(X_i) \mid X_j) \geq 0 \tag{16}$$

Therefore,

$$I(X_i; f(X_i)) \geq I(X_j; f(X_i)) \tag{17}$$

$\square$

**Lemma A.2.** Let $X = [X_1; X_2; ...; X_n]$ be a sequence of random variables, and $T = [T_1; T_2; ...; T_n] = [f(X_1); f(X_2); ...; f(X_n)]$ be a sequence of random variables generated by a deterministic function $f$. Then we have

$$I(X; T) \leq n \sum_{i=1}^{n} I(X_i; T_i) \tag{18}$$

*Proof.* Since $X = [X_1; X_2; ...; X_n]$ and $T = [T_1; T_2; ...; T_n]$ are language tokens with its corresponding local representations, we have

$$I(X; T) = I(X; T_1, T_2, ..., T_n) = \sum_{i=1}^{n} [H(T_i \mid T_1, T_2, ..., T_{i-1}) - H(T_i \mid X, T_1, T_2, ..., T_{i-1})] \tag{19}$$

$$\leq \sum_{i=1}^{n} [H(T_i) - H(T_i \mid X)] = \sum_{i=1}^{n} I(X; T_i) \tag{20}$$

$$\leq \sum_{i=1}^{n} \sum_{j=1}^{n} I(X_j; T_i) \leq n \sum_{i=1}^{n} I(X_i; T_i), \tag{21}$$

where the first inequality follows because conditioning reduces entropy, and the last inequality is because $I(X_i; T_i) \geq I(X_j; T_i)$ based on Lemma A.1. $\square$

Then we directly plug Lemma A.2 into Theorem 3.1, we have the lower bound of $\mathcal{L}_{\text{IB}}$ as

$$I(Y; T) - \beta I(X; T) \geq I(Y; T) - n\beta \sum_{i=1}^{n} I(X_i; T_i). \tag{22}$$

#### A.3.2 PROOF OF THEOREM 3.2

We first state an easily proven lemma,

**Lemma A.3.** For any $a, b \in [0, 1]$,

$$|a \log(a) - b \log(b)| \leq \phi(|a - b|), \tag{23}$$

where $\phi(\cdot) : \mathbb{R}_+ \to \mathbb{R}_+$ is defined as

$$\phi(x) = \begin{cases} 0 & x = 0 \\ x \log(\frac{1}{x}) & 0 < x < \frac{1}{e} \\ \frac{1}{e} & x > \frac{1}{e} \end{cases}. \tag{24}$$

It is easy to verify that $\phi(x)$ is a continuous, monotonically increasing, concave and subadditive function.

Now, we can proceed with the proof of Theorem 3.2.

*Proof.* We use the fact that

$$|I(Y;T) - I(Y;T')| \leq |H(T \mid Y) - H(T' \mid Y)| + |H(T) - H(T')| \tag{25}$$

and bound each of the summands on the right separately.

We can bound the first summand as follows:

$$|H(T \mid Y) - H(T' \mid Y)| \leq \sum_y p(y)|H(T \mid Y = y) - H(T' \mid Y = y)| \tag{26}$$

$$= \sum_y p(y)|\sum_t p(t \mid y) \log(1/p(t \mid y)) - \sum_t q(t \mid y) \log(1/q(t \mid y))| \tag{27}$$

$$\leq \sum_y p(y) \sum_t |p(t \mid y) \log p(t \mid y) - q(t \mid y) \log q(t \mid y)| \tag{28}$$

$$\leq \sum_y p(y) \sum_t \phi(|p(t \mid y) - q(t \mid y)|) \tag{29}$$

$$= \sum_y p(y) \sum_t \phi(|\sum_x p(t \mid x)[p(x \mid y) - q(x \mid y)]|), \tag{30}$$

where

$$p(x \mid y) = \frac{p(y \mid x)p(x)}{\sum_x p(y \mid x)p(x)} \tag{31}$$

$$q(x \mid y) = \frac{p(y \mid x)q(x)}{\sum_x p(y \mid x)q(x)}. \tag{32}$$

Since $\sum_{x \in \mathcal{X} \cup \mathcal{X}'} p(x \mid y) - q(x \mid y) = 0$ for any $y \in \mathcal{Y}$, we have that for any scalar $a$,

$$|\sum_x p(t \mid x)[p(x \mid y) - q(x \mid y)])| \tag{33}$$

$$= |\sum_x (p(t \mid x) - a)(p(x \mid y) - q(x \mid y))| \tag{34}$$

$$\leq \sqrt{\sum_x (p(t \mid x) - a)^2} \sqrt{\sum_x (p(x \mid y) - q(x \mid y))^2}. \tag{35}$$

Setting $a = \frac{1}{|\mathcal{X} - \mathcal{X}'|} \sum_{x \in \mathcal{X} \cup \mathcal{X}'} p(t \mid x)$ we get

$$|H(T \mid Y) - H(T' \mid Y)| \leq \sum_y p(y) \sum_t \phi(\sqrt{V(\mathbf{p}(t \mid x \in \mathcal{X} \cup \mathcal{X}'))} \cdot ||\mathbf{p}(x \mid y) - \mathbf{q}(x \mid y)||_2), \tag{36}$$

where for any real-value vector $\mathbf{a} = (a_1, ..., a_n)$, $V(\mathbf{a})$ is defined to be proportional to the variance of elements of $\mathbf{a}$:

$$V(\mathbf{a}) = \sum_{i=1}^{n}(a_i - \frac{1}{n}\sum_{j=1}^{n}a_j)^2, \tag{37}$$

$\mathbf{p}(t \mid x \in \mathcal{X} \cup \mathcal{X}')$ stands for the vector in which entries are $p(t \mid x)$ with different values of $x \in \mathcal{X} \cup \mathcal{X}'$ for a fixed $t$, and $\mathbf{p}(x \mid y)$ and $\mathbf{q}(x \mid y)$ are the vectors in which entries are $p(x \mid y)$ and $q(x \mid y)$, respectively, with different values of $x \in \mathcal{X} \cup \mathcal{X}'$ for a fixed $y$.

Since

$$||\mathbf{p}(x \mid y) - \mathbf{q}(x \mid y)||_2 \le ||\mathbf{p}(x \mid y) - \mathbf{q}(x \mid y)||_1 \le 2, \tag{38}$$

it follows that

$$|H(T \mid Y) - H(T' \mid Y)| \le \sum_y p(y) \sum_t \phi\big(2\sqrt{V(\mathbf{p}(t \mid x \in \mathcal{X} \cup \mathcal{X}'))}\big) \tag{39}$$

Moreover, we have

$$\sqrt{V(\mathbf{p}(t \mid x \in \mathcal{X} \cup \mathcal{X}'))} \le \sqrt{V(\mathbf{p}(t \mid x \in \mathcal{X})) + V(\mathbf{p}(t \mid x \in \mathcal{X}'))} \tag{40}$$
$$\le \sqrt{V(\mathbf{p}(t \mid x \in \mathcal{X}))} + \sqrt{V(\mathbf{p}(t \mid x \in \mathcal{X}'))}, \tag{41}$$

where the first inequality is because sample mean is the minimizer of the sum of the squared distances to each sample and the second inequality is due to the subadditivity of the square root function. Using the fact that $\phi(\cdot)$ is monotonically increasing and subadditive, we get

$$|H(T \mid Y) - H(T' \mid Y)| \le \sum_y p(y) \sum_t \phi\big(2\sqrt{V(\mathbf{p}(t \mid x \in \mathcal{X}))}\big)$$
$$+ \sum_y p(y) \sum_t \phi\big(2\sqrt{V(\mathbf{p}(t \mid x \in \mathcal{X}'))}\big) \tag{42}$$

Now we explicate the process for establishing the bound for $\sum_y p(y) \sum_t \phi\big(2\sqrt{V(\mathbf{p}(t \mid x \in \mathcal{X}))}\big)$ and the one for $\sum_y p(y) \sum_t \phi\big(2\sqrt{V(\mathbf{p}(t \mid x \in \mathcal{X}'))}\big)$ can be similarly derived.

By definition of $V(\cdot)$ and using Bayes' theorem $p(t \mid x) = \frac{p(t)p(x|t)}{p(x)}$ for $x \in \mathcal{X}$, we have that

$$\sqrt{V(\mathbf{p}(t \mid x \in \mathcal{X}))} = p(t)\sqrt{\sum_{x \in \mathcal{X}}\big(\frac{p(x \mid t)}{p(x)} - \frac{1}{|\mathcal{X}|}\sum_{x' \in \mathcal{X}}\frac{p(x' \mid t)}{p(x')}\big)^2} \tag{43}$$

Denoting $\mathbf{1} = (1, ..., 1)$, we have by the triangle inequality that

$$\sqrt{\sum_{x \in \mathcal{X}} \left(\frac{p(x \mid t)}{p(x)} - \frac{1}{|\mathcal{X}|} \sum_{x' \in \mathcal{X}} \frac{p(x' \mid t)}{p(x')}\right)^2} \tag{44}$$

$$\leq ||\frac{\mathbf{p}(x \mid t)}{\mathbf{p}(x)} - \mathbf{1}||_2 + \sqrt{\sum_{x \in \mathcal{X}} \left(1 - \frac{1}{|\mathcal{X}|} \sum_{x' \in \mathcal{X}} \frac{p(x' \mid t)}{p(x')}\right)^2} \tag{45}$$

$$= ||\frac{\mathbf{p}(x \mid t)}{\mathbf{p}(x)} - \mathbf{1}||_2 + \sqrt{|\mathcal{X}|\left(1 - \frac{1}{|\mathcal{X}|} \sum_{x' \in \mathcal{X}} \frac{p(x' \mid t)}{p(x')}\right)^2} \tag{46}$$

$$= ||\frac{\mathbf{p}(x \mid t)}{\mathbf{p}(x)} - \mathbf{1}||_2 + \sqrt{\frac{1}{|\mathcal{X}|}\left(|\mathcal{X}| - \sum_{x' \in \mathcal{X}} \frac{p(x' \mid t)}{p(x')}\right)^2} \tag{47}$$

$$= ||\frac{\mathbf{p}(x \mid t)}{\mathbf{p}(x)} - \mathbf{1}||_2 + \frac{1}{\sqrt{|\mathcal{X}|}}\left|\sum_{x' \in \mathcal{X}} \left(1 - \frac{p(x' \mid t)}{p(x')}\right)\right| \tag{48}$$

$$\leq ||\frac{\mathbf{p}(x \mid t)}{\mathbf{p}(x)} - \mathbf{1}||_2 + \frac{1}{\sqrt{|\mathcal{X}|}}||\frac{\mathbf{p}(x \mid t)}{\mathbf{p}(x)} - \mathbf{1}||_1 \tag{49}$$

$$\leq \left(1 + \frac{1}{\sqrt{|\mathcal{X}|}}\right)||\frac{\mathbf{p}(x \mid t)}{\mathbf{p}(x)} - \mathbf{1}||_1 \tag{50}$$

$$\leq \frac{2}{\min_{x \in \mathcal{X}} p(x)}||\mathbf{p}(x \mid t) - \mathbf{p}(x)||_1 \tag{51}$$

From an inequality linking KL-divergence and the $l_1$ norm, we have that

$$||\mathbf{p}(x \mid t) - \mathbf{p}(x)||_1 \leq \sqrt{2\log(2)D_{\text{KL}}[\mathbf{p}(x \mid t)||\mathbf{p}(x)]} \tag{52}$$

Plugging Eq. (52) into Eq. (51) and using Eq. (43), we have the following bound:

$$\sqrt{V(\mathbf{p}(t \mid x \in \mathcal{X}))} \leq \frac{B}{2}p(t)\sqrt{d_t}, \tag{53}$$

where $B = \frac{4\sqrt{2\log(2)}}{\min_{x \in \mathcal{X}} p(x)}$ and $d_t = D_{\text{KL}}[\mathbf{p}(x \mid t)||\mathbf{p}(x)]$.

We will first proceed the proof under the assumption that $Bp(t)\sqrt{d_t} \leq \frac{1}{e}$ for any $t$. We will later see that this condition can be discarded. If $Bp(t)\sqrt{d_t} \leq \frac{1}{e}$, then

$$\sum_t \phi\left(2\sqrt{V(\mathbf{p}(t \mid x \in \mathcal{X}))}\right) \tag{54}$$

$$\leq \sum_t Bp(t)\sqrt{d_t}\left(\log(\frac{1}{B}) + \log(\frac{1}{p(t)d_t})\right) \tag{55}$$

$$= B\log(\frac{1}{B})\sum_t p(t)\sqrt{d_t} + B\sum_t p(t)\sqrt{d_t}\log(\frac{1}{p(t)d_t}) \tag{56}$$

$$\leq B\log(\frac{1}{B})||\mathbf{p}(t)\sqrt{\mathbf{d}_t}||_1 + B||\sqrt{\mathbf{p}(t)\sqrt{\mathbf{d}_t}}||_1, \tag{57}$$

where the last inequality is due to an easily proven fact that for any $x > 0$, $x\log(\frac{1}{x}) \leq \sqrt{x}$. We $\mathbf{p}(t)$ and $\mathbf{d}(t)$ are vectors comprising $p(t)$ and $d_t$ with different values of $t$, respectively.

Using the following two inequalities:

$$||\mathbf{p}(t)\sqrt{\mathbf{d}_t}||_1 \leq \sqrt{|\mathcal{T}|}||\mathbf{p}(t)\sqrt{\mathbf{d}_t}||_2 \leq \sqrt{|\mathcal{T}|}||\sqrt{\mathbf{p}(t)\mathbf{d}_t}||_2 \tag{58}$$

and

$$||\sqrt{\mathbf{p}(t)\sqrt{\mathbf{d}_t}}||_1 \leq \sqrt{|\mathcal{T}|}||\sqrt{\mathbf{p}(t)\sqrt{\mathbf{d}_t}}||_2 \tag{59}$$

$$= \sqrt{\mathcal{T}}\sqrt{||\mathbf{p}(t)\sqrt{\mathbf{d}_t}||_1} \leq |\mathcal{T}|^{3/4}\sqrt{||\sqrt{\mathbf{p}(t)\mathbf{d}_t}||_2} \tag{60}$$

we have

$$\sum_t \phi\left(2\sqrt{V(\mathbf{p}(t \mid x \in \mathcal{X}))}\right) \leq B \log(\frac{1}{B})\sqrt{|\mathcal{T}|}||\sqrt{\mathbf{p}(t)\mathbf{d}_t}||_2 + B|\mathcal{T}|^{3/4}\sqrt{||\sqrt{\mathbf{p}(t)\mathbf{d}_t}||_2}. \quad (61)$$

Using the equality

$$||\sqrt{\mathbf{p}(t)\mathbf{d}_t}||_2 = \sqrt{\mathbb{E}[D_{\mathrm{KL}[p(x|t)||p(x)]}]} = \sqrt{I(X;T)}, \quad (62)$$

we reach the following bound

$$\sum_t \phi\left(2\sqrt{V(\mathbf{p}(t \mid x \in \mathcal{X}))}\right) \quad (63)$$

$$\leq B \log(\frac{1}{B})|\mathcal{T}|^{1/2}I(X;T)^{1/2} + B|\mathcal{T}|^{3/4}I(X;T)^{1/4}. \quad (64)$$

Plug Lemma A.2 into the equation above, we have

$$\sum_t \phi\left(2\sqrt{V(\mathbf{p}(t \mid x \in \mathcal{X}))}\right) \quad (65)$$

$$\leq B \log(\frac{1}{B})|\mathcal{T}|^{1/2}(n\sum_{i=1}^{n} I(X_i;T_i))^{1/2} + B|\mathcal{T}|^{3/4}(n\sum_{i=1}^{n} I(X_i;T_i))^{1/4} \quad (66)$$

$$\leq \sqrt{n}B \log(\frac{1}{B})|\mathcal{T}|^{1/2}\sum_{i=1}^{n} I(X_i;T_i)^{1/2} + n^{1/4}B|\mathcal{T}|^{3/4}\sum_{i=1}^{n} I(X_i;T_i)^{1/4} \quad (67)$$

We now show the bound is trivial if the assumption that $Bp(t)\sqrt{d_t} \leq \frac{1}{e}$ does not hold. If the assumption does not hold, then there exists a $t$ such that $Bp(t)\sqrt{d_t} > \frac{1}{e}$. Since

$$\sqrt{I(X;T)} = \sqrt{\sum_t p(t)d_t} \geq \sum_t p(t)\sqrt{d_t} \geq p(t)\sqrt{d_t} \quad (68)$$

for any t, we get that $\sqrt{I(X;T)} \geq \frac{1}{eB}$. Since $|\mathcal{T}| \geq 1$ and $C \geq 0$, we get that our bound in Eq. (63) is at least

$$B \log(\frac{1}{B})|\mathcal{T}|^{1/2}I(X;T)^{1/2} + B|\mathcal{T}|^{3/4}I(X;T)^{1/4} \quad (69)$$

$$\geq \sqrt{|\mathcal{T}|}(\frac{\log(1/B)}{e} + \frac{B^{1/2}|\mathcal{T}|^{1/4}}{e^{1/2}}) \quad (70)$$

Let $f(c) = \frac{\log(1/c)}{e} + \frac{c^{1/2}|\mathcal{T}|^{1/4}}{e^{1/2}}$. It can be verifed that $f'(c) > 0$ if $c > 0$. Since $B > 4\sqrt{2\log(2)}$ by the definition of $B$, we have $f(B) > f(4\sqrt{2\log(2)}) > 0.746$. Therefore, we have

$$B \log(\frac{1}{B})|\mathcal{T}|^{1/2}I(X;T)^{1/2} + B|\mathcal{T}|^{3/4}I(X;T)^{1/4} \quad (71)$$

$$\geq 0.746\sqrt{|\mathcal{T}|} \geq \log(|\mathcal{T}|) \quad (72)$$

Therefore, if indeed $Bp(t)\sqrt{d_t} > \frac{1}{e}$ for some $t$, then the bound in Eq. (63) is trivially true, since $H(T \mid Y)$ is within $[0, \log(|\mathcal{T}|)]$. Similarly, we can establish a bound for $\sum_t \phi\left(2\sqrt{V(\mathbf{p}(t \mid x \in \mathcal{X}'))}\right)$ as follows:

$$\sum_t \phi\left(2\sqrt{V(\mathbf{p}(t \mid x \in \mathcal{X}'))}\right) \leq \sqrt{n}B' \log(\frac{1}{B'})|\mathcal{T}|^{1/2}\sum_{i=1}^{n} I(X_i';T_i')^{1/2} + n^{1/4}B'|\mathcal{T}|^{3/4}\sum_{i=1}^{n} I(X_i';T_i')^{1/4}, \quad (73)$$

where $B' = \frac{4\sqrt{2\log(2)}}{\min_{x \in \mathcal{X}'} q(x)}$.

Plugging Eq. (73) and Eq. (65) into Eq. (42), we get

$$
|H(T \mid Y) - H(T' \mid Y)| \leq \sqrt{n} B \log(\frac{1}{B})|\mathcal{T}|^{1/2} \sum_{i=1}^{n} I(X_i; T_i)^{1/2} + n^{1/4} B |\mathcal{T}|^{3/4} \sum_{i=1}^{n} I(X_i; T_i)^{1/4} +
$$
$$
\sqrt{n} B' \log(\frac{1}{B'})|\mathcal{T}|^{1/2} \sum_{i=1}^{n} I(X_i'; T_i')^{1/2} + n^{1/4} B' |\mathcal{T}|^{3/4} \sum_{i=1}^{n} I(X_i'; T_i')^{1/4}
$$
$$(74)$$

Now we turn to the third summand in Eq. (25), we have to bound $|H(T) - H(T')|$.

Recall the definition of $\epsilon$-bounded adversarial example. We denote the set of the benign data representation $t$ that are within the $\epsilon$-ball of $t'$ by $Q(t')$. Then for any $t \in Q(t')$, we have

$$
||t_i' - t_i|| \leq \epsilon, \tag{75}
$$

for $i = 1, 2, ..., n$. We also denote the number of the $\epsilon$-bounded adversarial examples around the benign representation $t$ by $c(t)$. Then we have the distribution of adversarial representation $t'$ as follows:

$$
q(t') = \sum_{t \in Q(t')} \frac{p(t)}{c(t)} \tag{76}
$$

$$
|H(T) - H(T')| \tag{77}
$$
$$
= |\sum_t p(t) \log p(t) - \sum_{t'} q(t') \log q(t')| \tag{78}
$$
$$
= |\sum_t p(t) \log p(t) - \sum_{t'} \left[ \left( \sum_{t \in Q(t')} \frac{p(t)}{c(t)} \right) \log \left( \sum_{t \in Q(t')} \frac{p(t)}{c(t)} \right) \right]| \tag{79}
$$
$$
\leq |\sum_t p(t) \log p(t) - \sum_{t'} \sum_{t \in Q(t')} \frac{p(t)}{c(t)} \log(\frac{p(t)}{c(t)})| \tag{80}
$$
$$
= |\sum_t p(t) \log p(t) - \sum_t c(t) \frac{p(t)}{c(t)} \log(\frac{p(t)}{c(t)})| \tag{81}
$$
$$
= |\sum_t p(t) \log c(t)|, \tag{82}
$$

where the inequality is by log sum inequality. If we denote the $C = \max_t c(t)$ which is the maximum number of $\epsilon$-bounded textual adversarial examples given a benign representation $t$ of a word sequence $x$, we have

$$
|H(T) - H(T')| \tag{83}
$$
$$
\leq |\sum_t p(t) \log c(t)| \tag{84}
$$
$$
\leq |\sum_t p(t) \log C| = \log C. \tag{85}
$$

Note that given a word sequence $x$ of $n$ with representation $t$, the number of $\epsilon$-bounded textual adversarial examples $c(t)$ is finite given a finite vocabulary size. Therefore, if each word has at most $k$ candidate word perturbations, then $\log C \leq n \log k$ can be viewed as some constants depending only on $n$ and $\epsilon$.

Now, combining Eq. (25), Eq. (74) and Eq. (85), we prove the bound in Theorem 3.2. □

