# OpenReview forum: "InfoBERT: Improving Robustness of Language Models from An Information Theoretic Perspective"
_ICLR.cc/2021/Conference — ICLR 2021 Poster_

### Official Review · AnonReviewer1 · 2020-10-27
**Weak accept**

**Rating:** 6
**Confidence:** 3

**Review:**

This work (InfoBERT) proposes additional objectives for transformer finetuning to obtain models more robust to adversarial inputs. The authors first propose a mutual information based information bottleneck objective, next the authors propose an adversarial loss inspired method for identifying robust features and a subsequent objective to emphasize the mutual information between global representations and these robust features. The experiments demonstrate that InfoBERT consistently outperforms other adversarial training approaches on a variety of adversarial evaluations.

I largely follow the logic behind the derivation, however I find some of the details unclear. I would like to see proofs for the theorems as well as an explanation of the assumptions under which the theorems hold. The experimental results are convincing, however there are no ablation studies to disentangle the performance contributions of the two proposed objectives. For the first point, the questions I have are as follows:

for equations 1-3, I find the integral notation to be a bit odd - isn't it common practice to put the dydt at the very end of the integral? Also you should consider leaving out punctuations from equations
for equation 5, why is there another 1/N inside the square brackets?
why is equation 7 true in the general case? Suppose n=1, is this essentially saying that any sample from an empirical distribution would provide a lower bound for the true distribution?
I'd like to see a proof for Theorems 3.1 and 3.2
how do the authors define stability and robustness? The manuscript talk about them in vague terms and they do not seem to be precisely defined
how does equation 9 follow from 6? Can you put in the intermediate steps? Also in this case what is N and what is M? And what happened to the multiplier n from equation 6?

---

> ### Author Response · Authors · 2020-11-13
> **Response to Reviewer #1**
>
> Thank you for your valuable comments.
>
> **Q1:** “disentangle the performance contributions of the two proposed objectives”
>
> **A1:** We investigate the independent impact of the two regularizers via ablation studies in Appendix A.2.1 (IB regularizer) and A.2.2 (Anchored Feature Regularizer). To make it more clear, we have added another section A.2.3 (Ablation Study on Disentangling Two Regularizers, updated now in OpenReview). In Table 8, we show that both regularizers can individually improve the robust accuracy on top of FreeLB with a similar margin. After combining both regularizers, InfoBERT achieves the best robust accuracy. We will make it more clear in the revision.
>
>
> **Q2:** “isn't it common practice to put the $dy \, dt$ at the very end of the integral?”
>
> **A2:** We are following the practice of [1] for the lower bound of $I(Y;T)$. We have followed your suggestion and updated the equations in OpenReview. Thanks for pointing it out.
>
>
> **Q3:** “leaving out punctuations from equations for equation 5”
>
> **A3:** Thanks for the careful reading. Equation (5) is not the end of the sentence, but followed by a short sentence “with data samples  $\\{ {x}^{(i)}, {y}^{(i)} \\}_{i=1}^{N}$.”
>
>
> **Q4:** “another 1/N inside the square brackets”
>
> **A4:** For the upper bound of $I(X;T)$, we follow the derivation from [2]. Specifically, Eq. (4) is estimated with data samples $\\{x^{(i)}, y^{(i)}\\}$ based on Eq. (15) in [2], where we calculate the conditional distribution via positive pair $\\{x^{(i)}, t^{(i)}\\}$ and negative pairs $\\{x^{(i)}, t^{(j)} \\}$ (for $j=1$ to $N$).
>
>
> **Q5:** “Equation (7)”
>
> **A5:** Yes, you are right. In particular, when $n=1, X = [X_1],  I(X; T) = \sum_{i=1}^{n} I(X_i; T_i)$. Therefore, LHS=RHS, and it provides the lower bound for Equation (1).
>
>
>
> **Q6:** “Define stability and robustness”
>
> **A6:** We define ‘stability’ as the stability of feature representation for local words before and after potential perturbation. By Algorithm 1, we ensure that local anchored features are perturbed under the threshold $c_h$. Here theorem 3.1 and 3.2 only focus on robustness.
>
> The definition of robustness is the performance on the adversarial input, and we try to improve robustness by diminishing the performance gap between adversarial input and benign input. Since $I(Y;T)$ and $I(Y;T’)$ can be regarded as model performance on benign and adversarial data ([1][2]), we bound the performance gap $||I(Y;T) - I(Y;T’)||$ by $I(X;T)$ and $I(X’; T’)$, and formalize that the performance gap will become closer when $I(X;T)$ and $I(X’; T’)$ get closer, thus leading to better adversarial robustness.
>
>
> **Q7:** “Eq (9)”
>
> **A7:** N is the size of the dataset. M is the number of local anchored features. Sorry for the typos in our previous version. We have updated it, and have added the multiplier $n$ back.
>
>
> **Q8:** “Intermediate steps from Eq (6) to Eq (9)”
>
> **A8:** Yes, you are right that Eq (6) is followed by Eq (9). In particular, we add the anchored feature regularizer (the $\alpha$ term) to Eq (6). The anchored feature regularizer tries to align local stable features $T_{k_j}$ found by Algorithm 1 with sentence-level global representation $Z$, and increase MI $I(T_{k_j}; Z)$ between them. Therefore, we add this term from previous objective function (Eq (6)) to Eq (9) and add the coefficient \alpha to control the trade-off.
>
> [1] Alemi, A.A., Fischer, I.S., Dillon, J.V., & Murphy, K. Deep Variational Information Bottleneck. (ICLR 2017)
>
> [2] Cheng et al. CLUB: A Contrastive Log-ratio Upper Bound of Mutual Information. (ICML 2020)

---

> ### Comment · AnonReviewer1 · 2020-11-22
> **Thank you for the response**
>
> Thank you authors for the response. I maintain my original score. RE the point "Suppose n=1, is this essentially saying that any sample from an empirical distribution would provide a lower bound for the true distribution?"
>
> I am still unclear why this would be the case in general. My understanding of the authors' response is that all examples provide a lower bound for the true distribution. Doesn't this mean that they all evaluate to the same quantity? In general I stand by my original comment that this paper needs more TLC in terms of its writing. In particular, the proofs need to be elaborated upon. It would have been nice to see these changes during the rebuttal.

---

> > ### Author Response · Authors · 2020-11-24
> > **Thank you for the follow-up comments**
> >
> > Thank you for the valuable follow-up comments.
> >
> > **Q1**: “why is equation 7 true in the general case? Suppose n=1, is this essentially saying that any sample from an empirical distribution would provide a lower bound for the true distribution”
> >
> > **A1**: Thank you for the follow-up question. We are sorry for making the confusion. Note that $X_i$ does not refer to the training examples in the dataset to represent true distribution. As discussed in the paragraph above Equation (7), $n$ is sequence length and $X=[X_1, X_2, …, …, X_n]$ representing input sentences with $X_i$ being word token at $i$-th index. Here, theorem 3.1 proposes a feasible way to calculate $I(X;T)$: when $n$ is large, $X$ and $T$ are high-dimensional features that are difficult to compute $I(X;T)$, therefore we can instead optimize the lower bound of $I(X;T)$ by factoring $X$ and $T$ into their localized representation $X_i$ and $T_i$, and calculate $n\beta \sum_{i=1}^{n} I(X_i; T_i)$ that are easier to compute.
> >
> > **Q2**: “the proofs need to be elaborated upon”
> >
> > **A2**: The intuition of theorem 3.1 is discussed in the previous answer. Theorem 3.2 wants to bound the adversarial performance drop $| I(Y;T) - I(Y; T') |$ by the mutual information between X and T. Specifically, we use equation (19), where the conditional entropy $|H(T \mid Y) - H(T' \mid Y)|$ can be bound by Eq. (36) in terms of MI between benign/adversarial input and representation $I(X_i;T_i)$ and $I(X'_i;T'_i)$, and the second summand $|H(T) - H(T')|$ is upper-bounded by a constant (Eq. (79)). The details of both proofs can be found in Appendix A.3. We have made the illustration of theorems more clear in the revision.
> >
> > We have updated a new version to OpenReview. Please let us know if it solves the concern. Thank you!

---

### Official Review · AnonReviewer3 · 2020-10-28
**InfoBERT review**

**Rating:** 8
**Confidence:** 2

**Review:**


##########################################################################

Summary:

The paper proposes a novel learning framework for robust (against adversarial attacks) fine-tuning of pre-trained language models, that is based on information theoretic arguments. It introduces two regularization mechanisms and investigates their efficacy on various tasks.

##########################################################################

Reasons for score:

Overall I vote for accept. The approach is novel, interesting and well presented. The theoretical results seem to be sound. It also seems to outperform competitors in the field of adversarial language models. Concerning the experiments some questions remain, but I hope that the authors will address them in the rebuttal.

##########################################################################
Pros:

1. The idea is interesting and well formulated. The theoretical results seem to be correct to me.

2. The approach is tested on several standard datasets used in adversarial language models. It seems to outperform previous approaches.

3. The paper is well written and clearly structured


##########################################################################

Cons:


1. In my view the experiments seem to show a tendency towards a slightly worse performance on the more difficult tasks in comparison to the competitor methods. Thus, the better overall performance on the ANLI data could be driven by the easier tasks.


2. I couldn't find a clear description of the "global representation" Z. A more explicit description would be helpful.


##########################################################################

Questions during rebuttal period:


Please address and clarify the cons above


#########################################################################

Minor comments:

1. Is definition 3.1. a standard definition or is it introduced by the authors?

2. Page 4, definition 3 contains an incomplete sentence ("The q(x')....").

3. Page 6, "Evaluation Metrics": it should be stated witch argument is maximized.

4. Page 16, Lemma A1: in the Proof of the lemma I think that all instances of Y_i should be replaced with T_i. In formula (13), in the rightmost term the token index n should be i-1.

5. Page 17, formula (33): H(Y|Y) should probably be H(T|Y). Same goes for equ. (36).

6. Squares might be missing in formulas (37) to (43).

7. A reference to formula (44) would be nice.

---

> ### Author Response · Authors · 2020-11-13
> **Response to Reviewer #3**
>
> Thank you for your careful reading and comments.
>
> **Q1:** “slightly worse performance on the more difficult tasks”
>
> **A1:** We thank the reviewer for the insightful observation. To diversify the tasks, we conduct a comprehensive evaluation on common NLP tasks, including NLI and QA. We observe an overall improvement in our experiments. Specifically,  InfoBERT with adversarial training can achieve 5-10% robust accuracy improvement for ANLI, and around 6-point F1 score improvement for QA over the baseline. We do notice that the margin on TextFooler is relatively smaller (1-5%) and it could be potentially due to the fact that adversarial training has also improved the model against such specific attacks, therefore the final improvement can be marginal (1-5%). But if we compare with the vanilla baseline without adversarial training, the improvement is still significant (5-10%), which indicates that adversarial training is very effective in defending against adversarial attacks such as TextFooler, while it is not the case for other adversarial datasets. We will add this discussion in our revision.
>
>
> **Q2:** “global representation Z”
>
> **A2:** Thanks for pointing this out. BERT uses a final-layer hidden vector corresponding to the first input token ([CLS]) as sentence global representation (Devlin, J. et al. “BERT: Pre-training of Deep Bidirectional Transformers for Language Understanding.” NAACL-HLT (2019)). We follow the practice of (Delvin, et al. 2019) and use the final-layer CLS embedding to represent global sentence-level features. We will make it more clear in the revision.
>
> **Q3:** “definition 3.1”
>
> **A3:** Our definition is based on (Jacobsen, J. et al. “Excessive Invariance Causes Adversarial Vulnerability.”(2019)). Due to the discrete nature of text input space, we adapt the definition to NLP by measuring adversarial distortion in the token embedding space. We will make it clear in our revision.
>
>
> **Q4:** “it should be stated which argument is maximized”
>
> **A4:** Thanks for pointing it out. $\arg \max$ selects the class with the highest logits/probability. We have added the description in the revision (updated now in OpenReview).
>
>
> **Q5:** “comments 4-7”
>
> **A5:** We sincerely thank the reviewer for the careful reading, and we have corrected our typos in revision (updated now in OpenReview).

---

> > ### Comment · AnonReviewer3 · 2020-11-23
> > **Thank you for the response!**
> >
> > Thanks for the authors' response! I believe that my questions have been sufficiently answered. Furthermore, some errors/typos have already been corrected. I maintain my original score.

---

> > > ### Author Response · Authors · 2020-11-24
> > > **Thank you for the follow-up comments**
> > >
> > > Thank you very much for acknowledging our work as interesting and solid, as well as for the valuable careful reading.
> > >
> > > Your comments have substantially improved our manuscript. Thank you!

---

### Official Review · AnonReviewer4 · 2020-10-30
**not super convincing results. important ablations missing**

**Rating:** 4
**Confidence:** 4

**Review:**

## Summary:
The paper proposes two regularizers for finetuning pretrained mask LMs to improve the robustness of NLI and SQuAD models. When evaluated on adversarial NLI and SQuAD datasets, adding the regularizers to regular finetuning achieves a robust accuracy comparable to adversarial training baselines; adding the regularizers are added to adversarial training baselines archives extra robustness gains.

The first regularizer is an implementation of the Information Bottleneck principle specialized for contextual text representations. In the general IB objective, we seek to maximize the mutual information between the representation and the label, as well as minimizing the mutual information between the representation and the input. In this paper, the authors targets the token-level BERT representation. This design choice was not discussed in detail, but I assume it’s motivated by the observation that our models’ lack of robustness is often manifested in an overreliance on local, superficial features.

The second regularizer has a similar motivation. But instead of minimizing the mutual information between the input and the representation, the “anchoring feature” regularizer minimizes the mutual information between the global representation and the token level ones, specifically those that are “nonrobust and unuseful”. To identify “nonrobust and unuseful” tokens in each input, the authors use a heuristic based on input gradient norm, similar to how interpretability people generate heatmaps for text classification.

## Detailed comments
Interpretation of experimental results and choice of baselines: The abstract claims that “InfoBERT achieves state-of-the-art robust accuracy”. This is not accurate. The best numbers reported in this paper are achieved by applying InfoBert regularizers to FreeLB adversarial training. This can be seen as an ensemble of two (or three, if you count the two regularizers separately since they can be applied independent of each other) adversarial training methods. Ensembling usually helps robustness (see for example Tramèr et.al ICLR 2018 Ensemble Adversarial Training: Attacks and Defenses). For a fair comparison, FreeLB should be ensemble with another adversarial training method, or with FreeLB applied to a second model. When applied individually, the gain from InfoBERT has a much smaller advantage compared to the baselines.

Missing evaluations and ablations: An obvious ablation is missing: apply the two regularizers separately. I’m especially curious if both lead to gains on top of FreeLB. The paper has a sort of disproportionate treatment of the two regularizers. Both theorem 3.1 and 3.2 are talking about the IB regularizer, while a lot of design choices for the anchoring feature regularizer are proposed without justification or verification, e.g. the portion of useful and robust tokens. The anchoring feature regularizer relies on various heuristics (definition of usefulness and robustness), and if it turns out to be the main contributing factor to InfoBERT, it would be good to know - if others would like to apply InfoBERT on other tasks, they might need to tune these hyperparameters.
Formulation of the method: The authors cite the “localization” of the IB principle in the IB regularizer as part of the novelty of the method. However, this kind of localization can be found in e.g. Li & Eisner 2019: Specializing Word Embeddings (for Parsing) by Information Bottleneck (EMNLP), which is one of the first applications of IB principle to NLP with pretrained, contextualized representations. In the anchoring feature regularizer, the use of input gradient norm for filtering “nonrobust and unuseful” tokens is reminiscent of how interpretation methods generate saliency maps for text classification. Both are missing from the references.

Implications and verification of theorem 3.2: this is a minor point but in my opinion theorem 3.2 is a bit of an overkill just to solidify the intuition that “the performance gap becomes closer when I(X_i, T_i) and I(X’_i, T’_i) decreases”. It would be nice to verify empirically through experiment that this theorem is correct.

Finally, I encourage the authors to evaluate the method on more tasks and attacks, and especially focus on comparing against the naive adversarial training baseline. It would be good to have a better understanding of how much gain InforBERT brings and what are the most important factors.

---

> ### Author Response · Authors · 2020-11-13
> **Response to Reviewer #4**
>
> Thank you for your valuable comments.
>
> **Q1:** “targets the token-level (local) representation”
>
> **A1:** Thank you for pointing this out, and you are right that the lack of robustness due to local superficial features is one of the main motivations. In addition, we also discuss different design choices of adopting token-level (local) representation v.s. sentence-level (global) features via ablation experiments in Appendix A.2.1 Table 6. While both features can boost model robustness, we observe that using local features yields higher robust accuracy than global ones. We will make it more clear in the revision.
>
>
> **Q2:** “Interpretation of experimental results and choice of baselines”
>
> **A2:** Sorry for the confusion. We will make the comparison clear in the revision.
> First, we emphasize that InfoBERT is not ensembled with FreeLB. Different from (Tramèr et.al 2018), InfoBERT does not augment adversarial data, but serves as regularizers added during training without additional computational cost.
> Second, from theoretical perspective, Theorem 3.2 shows that reducing MI between *adversarial input* and representation $I(X’;T’)$ can lead to further improvement on robustness, thus applying InfoBERT to adversarial training is fair and standard in our evaluation setting. As a result, we indeed compare InfoBERT against other SOTA adversarial training baselines, such as SMART, which combines adversarial training and Bregman proximal point optimization as regularizers to improve robustness. We show that InfoBERT achieves the best robust accuracy among existing adversarial training methods. Besides, we show that even when applying InfoBERT individually, it achieves better or comparable performance than other baselines (Table 1 for RoBERTa and Table 4).
>
>
> **Q3:**  “Missing evaluations and ablations”
>
> **A3:** Detailed analysis of the individual impact of the two regularizers is provided in ablation studies in Appendix A.2.1 (IB regularizer) and A.2.2 (Anchored Feature Regularizer). To make it more clear, we have added another section A.2.3 (Ablation Study on Disentangling Two Regularizers, which is updated now in OpenReview). In Table 8, we show that both regularizers can individually improve robust accuracy on top of FreeLB with a similar margin. After combining both regularizers, InfoBERT achieves the best robust accuracy. We discuss the strategies for hyper-parameter tuning for both regularizers in Appendix A.1. We will make it more clear in the revision.
>
>
> **Q4:**  “Formulation of the method”
>
> **A4:** Our work mainly focuses on improving the robustness of language representations by formulating Information bottleneck as a regularizer, and our method aims to improve both local (word-level) representations and global (sentence-level) features; while Li & Eisner (2019) has a different formulation and setting, which focuses on designing a model (rather than a flexible regularizer) to extract information from local word representations.
> Thanks for mentioning the interpretation. We design the anchored features by observing the magnitude of textual attack perturbation in the embedding space. We also provide visualization of the “adversarial saliency map” in ablation study in Appendix 2.2 Table 7. We will discuss related work and analysis in the revision.
>
>
> **Q5:** “More tasks and more attacks”
>
> **A5:** Thanks for the suggestion. We indeed evaluated the proposed InfoBERT on existing common NLP tasks, comparing with SOTA baselines. Specifically, we choose two high-quality adversarial NLP datasets (ANLI and adversarial SQuAD) for different tasks (NLI and QA), plus one dataset created by effective textual adversarial attack TextFooler, and compare InfoBERT against SOTA adversarial training approaches including ALUM and SMART.
> We will add more evaluation against adversarial attacks in the future. Please let us know if there are further suggestions on the evaluation.
>
> We hope that you can raise your score if you find our answers address your questions. Thank you!

---

### Author Response · Authors · 2020-11-24
**General Response**

We thank all the reviewers for their time and valuable suggestions. Based on the reviews, we have corrected several typos and made the illustration more clear. We also include more experimental results.

Specifically, we made the following revisions:
1. We revised the equation typos and followed the common practice in terms of notations in Section 3.
2. We added more illustrations on our definition and theorems in Section 3.
3. We added more description of evaluation metrics in Section 4.
4. We made the ablation studies that we studied more clear in Section 4.
5. We fixed the typos in Appendix A.3.
6. Previously we investigated the impact of the two regularizers via ablation studies from Appendix A.2.1 (Ablation Study on IB regularizer) and A.2.2 (Ablation Study on Anchored Feature Regularizer). We now include another section A.2.3 (Ablation Study on Disentangling Two Regularizers) to compare the results and make the conclusion more clear. We show that both regularizers can individually improve the robust accuracy on top of FreeLB with a similar margin. After combining both regularizers, InfoBERT achieves the best robust accuracy.

All of our revisions are updated in OpenReview now and highlighted in blue. Thank you!

---

### Decision · Program_Chairs · 2021-01-07
**Final Decision**

**Decision:**

Accept (Poster)

**Comment:**

This paper introduces two regularizers that are meant to improve out-of-domain robustness when used in the fine-tuning of pretrained transformers like BERT. Results with ANLI and Adversarial SQuAD are encouraging.

Pros:
- New method with concrete improvements in several difficult task settings.
- New framing of adversarial generalization.

Cons:
- The ablations that are highlighted in the main paper body don't do a good job of isolating the specific new contributions. (Though the appendix provides enough detail that I'm satisfied that the main empirical contribution is sound.)
- Reviewers found the theoretical motivation very difficult to follow in places.